# Mapping of the Land Cover Changes in High Mountains of Western Carpathians between 1990–2018: Case Study of the Low Tatras National Park (Slovakia)

**Michaela Žoncová [1], Pavel Hronček [2] and Bohuslava Gregorová [1,*]**

[1] Department of Geography and Geology, Faculty of Natural Sciences, Matej Bel University in Banská Bystrica, Tajovského 40, 97401 Banská Bystrica, Slovakia; michaela.zoncova@umb.sk

[2] Department of Geo and Mining Tourism, Institute of Earth Resources, Faculty of Mining, Ecology, Process Control and Geotechnologies, Technical University of Kosice, Němcovej 32, 04001 Košice, Slovakia; pavel.hroncek@tuke.sk

**\*** Correspondence: bohuslava.gregorova@umb.sk; Tel.: +421-907-559-818

**Abstract:** At present, the protection of nature and landscape in the high mountains of the Western Carpathians, protected as national parks, is becoming increasingly at the forefront of society's interests in connection with the development of their economic use and the development of mass tourism. Our research was focused on analyzing the extent and character of land cover changes in the Low Tatras National Park in Slovakia over the last 30 years (1990–2018) using CORINE land cover (CLC) data. The period captures almost the entire existence of the Slovak Republic. Therefore, it was possible to evaluate the landscape changes in the protected area and to identify barriers and possibilities of its long-term sustainable development. Based on computer modeling, the main areas of the land cover changes were identified, and on the basis of historical-geographical and field research, land cover flows were determined and justified in the studied landscape of the national park. Changes were monitored using three methods: by comparing CLC maps over the years, by analyzing land cover flows, and by comparing landscape metrics obtained through the PatchAnalyst. Land cover changes occurred on up to 20% of the national park area in the given period. The most significant change was observed in the CLC class coniferous forests, with almost a 12% decrease. Conversely, there was an increase of more than 11% in the CLC class transitional woodland-shrub.

**Keywords:** CORINE land cover; mapping of changes; GIS tools; land cover flows; protected areas; Low Tatras National Park

## 1. Introduction

Land cover changes (whether natural or occurring by anthropogenically affected development) are a continuous process worldwide [1–5], especially in developing countries of Asia and South America, but also in the ("post-socialist") countries of Central and Eastern Europe, including Slovakia. Socio-political reforms occurred after 1989, and subsequent transformations that began after 2004, when Slovakia joined the European Union, can be considered as the leading causes of land cover changes in this region. Issues of environmental protection also came to the forefront, and in 2002 the Act No. 543/2002 Coll. on Nature and Landscape Protection was adopted. This act strengthened the protection of natural balance, the protection of the diversity of natural conditions and life forms, natural values and creates the preconditions for the sustainable use of natural

resources and ecosystem services, taking into account the economic, social and cultural needs as well as regional and local circumstances. The act defines five zones of protection, the fifth being the highest.

Large-scale protected areas, which include national parks and protected landscape areas, are declared in Slovakia within the scope of nature conservation. A national park is defined as an area of over 1000 ha, predominantly with ecosystems substantially unchanged by human activity or with a unique and natural landscape structure constituting supra-regional biocentres and the most significant natural heritage, in which nature protection and conservation are superior to other activities. Territories of national parks fall under the third zone of protection.

One prominent body of research on the mountain environment in Slovakia focuses on geodynamic and geomorphological processes [6,7]. The second prominent body of research focuses on changes in the landscape structure, but this research captures especially changes at the turn of the 18th, 19th and 20th centuries. The main processes in this period were mainly the development of urbanization and agriculture [8] or the transformation of scattered settlements into recreational areas [9]. Changes in the landscape structure are partly a reflection of social changes that occurred after 1949 and later after 1989 in Slovakia [10]. All of these processes resulted in changes in ecological stability, both in the mountain environment and in the lowlands [11,12]. In recent years, the disproportionate spatial expansion of recreational infrastructure [13–15] has had a significant impact on the landscape, and it is, therefore, essential to set the limit of unbearability and to develop tourism inside the protected areas only up to a point, where no disruption of elemental links between ecosystems can be guaranteed [16]. This phenomenon often becomes irreversible when humans disproportionately affect the natural environment, affecting water [17], soil [18], flora, fauna, and the overall biodiversity of the landscape [19,20] in the protected areas. Changing climate, frequent temperature fluctuations, fires [21], windstorms, torrential rains and storms, droughts, and other natural phenomena also affect changes in nature. The most important anthropogenic influences are grazing, intensification and extensification of agriculture, mining of raw materials, areal growth of rural settlements, the development of recreational infra, and suprastructure.

The monitoring of landscape changes in Europe is mostly conducted using the CORINE land cover (CLC) database [7,10,22–24], which is considered to be the most complex database of spatial-temporal data. The functioning of the CLC database is financed by the Member States of the European Union, and it is managed by the European Environmental Agency (EEA) and is one of the products of the Copernicus Land Monitoring Service (CLMS). The extent and quality of information within the CLC database are different across European countries. The CLC database of Slovakia is one of the most accurate and complete. The quality and detail result from the use of a minimum scale (1:50,000) as well as from the adaptation of its legend to the specific local conditions [25].

The main aim of our research is to evaluate land cover changes in the Low Tatras National Park between 1990 and 2018. The selection of time periods was primarily based on data availability. These data also capture significant historical context since social-political changes in 1989, accession to the European Union in 2004 to nowadays. We have specified three sub-objectives within our research:

1.  to find out the character of landscape structure based on a comparative analysis of land cover maps from 1990 and 2018 using the CORINE land cover (CLC) data;
2.  to assess land cover flows based on CORINE land cover change layers in period 1990–2000, 2000–2006, 2006–2012, 2012–2018;
3.  to describe landscape structure changes based upon landscape metrics calculations.

    The following primary research questions should be answered:

- What type of landscape changes prevailed in the Low Tatras National Park during the research period?
- What was the intensity of landscape changes in the observed period in the Low Tatras National Park?
- How do the land cover changes influence the ecological attributes of a national park?

- What were the causes and processes (drivers) of landscape changes in the Low Tatras National Park within the monitored period?

The research of land cover changes is necessary to identify the negative anthropogenically conditioned and created processes and phenomena in the protected landscape, their prediction, analysis, prevention, and elimination concerning the active management of the landscape.

## 2. Materials and Methods

### 2.1. Territory of Interest

The Low Tatras National Park is located in the central part of the Western Carpathians (Figure 1). It spreads in the top part of the Low Tatras Mountains, which is the second-highest in the Carpathian Arc. The asymmetric vault of the central mountain ridge, located in the center of Slovakia, is significantly extended in the west-east direction. The highest point of the mountain range is the summit of Ďumbier (2043 m). From the orographic point of view, the territory of the national park consists mainly of the beforementioned Low Tatras, while some parts of Veľká Fatra Mountains, Staré Hory Mountains, Zvolen Basin, Horehronské Podolie Basin, Podtatranská Basin, Kozie chrbty Mountains and Spiš-Gemer Karst extend partially to this area.

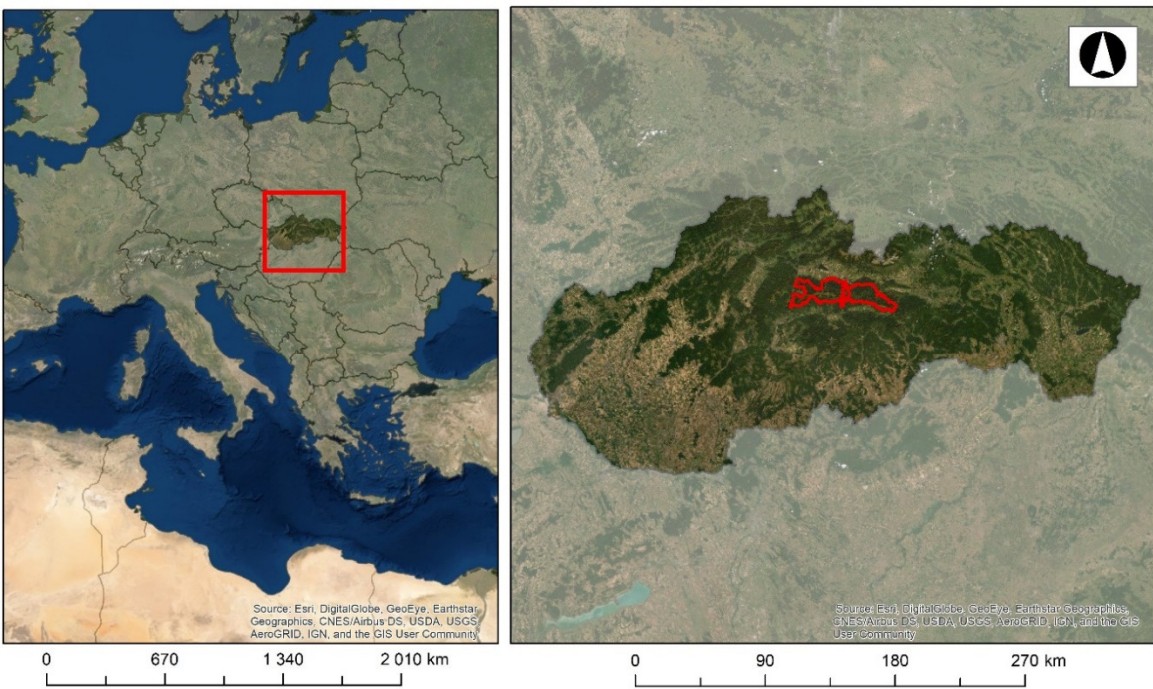

**Figure 1.** The location and demarcation of Low Tatras National Park in Slovakia.

The main ridge of the Low Tatras is built by the crystalline core (Tatrikum). It consists mainly of rocks formed in the Paleozoic when large rock complexes metamorphosed at the sea bottom during the younger geological times. During the long-term geological-tectonic development, these rocks were lifted and folded into a complex vaulted structure. The core was encapsulated initially by sandstone, shale, dolomite, and limestone strata, which were gradually removed by denudation, but mainly by the Quaternary glacial activity. The process has been particularly evident on the northern slopes of the mountain range, which is why the main range has a distinctly asymmetric profile (the northern slopes are much steeper and shorter than the southern ones) [26,27].

Glacial relief is characterized by steep rock walls, glacial cirques, moraines, and glacial lakes. There were 16 glaciers in the territory of the Low Tatras National Park. The longest one was in Dúbravská Valley and reached a length of more than 6000 m. The largest glacier lake ("pleso") is

Vrbické Lake with an original area of 0.73 ha, which is situated in Demänovská Valley and was created on the frontal moraine after the glacier retreat [28].

Up to 5 climatic-geographical types (the highland climate-very cold, cold, cool and moderately warm, and the basin climate-cold) can be distinguished in the Low Tatras due to high vertical articulation (from 500 to 2043 m a.s.l.). Mountain locations above 1500 m a.s.l. have a very cold mountain climate with average temperatures of −7 °C to −8 °C in January and approximately −9 °C on the main ridge [29,30].

The highest annual rainfall values are reached in the area between the peaks Prašivá and Dereše, which is caused by the global western streaming (1400–1700 mm on average, maximum annual totals are 1900–2300 mm, and the minimum annual rainfall is about 1000 mm) [31].

The streaming has a strongly variable direction due to local relief shapes. In the basins oriented in the west to east direction, the wind generally flows in the same direction, while the north-south streaming prevails in the mountain ridge part. However, winds from the northwest and southwest are also frequent. The least occurring are south-eastern and northeastern winds. The average wind speed increases with the increasing altitude reaching an average of 9.6 m/s at Chopok (2023 m) and only 1.2 m/s at Jasná (1200 m) [29].

Large areas of the National Park territory had been covered mostly by beech and beech-fir primeval forests until the 15th century, except for the peak areas of the highest part of Chopok and Ďumbier massifs. The vast extent of the mountain range and the articulation of the relief enabled primeval forests to retain their natural character for quite a long time. The gradual settlement, development of mining and metallurgy, and since the 13th century, the related logging and pasturing since the 14th century, have significantly accelerated the deforestation process [32]. A substantial part of today's national park was mostly a clear-cutting in the 16th century [33]. The current composition of forest vegetation was significantly affected by artificial restoration since the 19th century, when the emphasis on spruce monocultures began.

Heavily anthropogenically affected beech and fir-beech forests nowadays cover the western edges of the national park mountain range, lined with oak stands in contact with basins. A wide belt of monocultural spruce stands extends above this level, which completely prevails in the eastern part (Kráľovohoľské Tatras) and creates the timberline at an altitude of 1500–1600 m a.s.l. Forest pine grows on the rocky northern slopes. A dwarf mountain pine belt gradually changes into human-made mountain grasslands occurring from an altitude of about 1400 m a.s.l. [30,34].

The Low Tatras National Park was declared (as the third national park in Slovakia) by Decree of the Government of the SSR no. 119/1978 Coll [35]. Subsequently, the Ministry of Culture of the SSR issued Decree No. 120/1978 [36], establishing its status on 17 October 1978. It was confirmed by Act No. 287/1994 Coll. on Nature and Landscape Protection [37], as well as Act no. 543/2002 Coll [29,38,39]. The area of the largest national park in Slovakia was limited to 205,085 ha, including a protection zone (81,095 ha was the area of the national park itself, and 123,990 ha was the area of the protection zone). The border length was 340 km and up to 575 km together, including the boundaries of the protection zone [39]. There were eight small-scale specially protected areas at the time of its declaration, while currently there are 48 of them. There are also protected areas of NATURA 2000:2 sites—birds directive and 10 sites—habitats directive. Almost the whole area of the national park is under specific protection (except the part of the Demänová valley and Bystrá valley) (Figure 2).

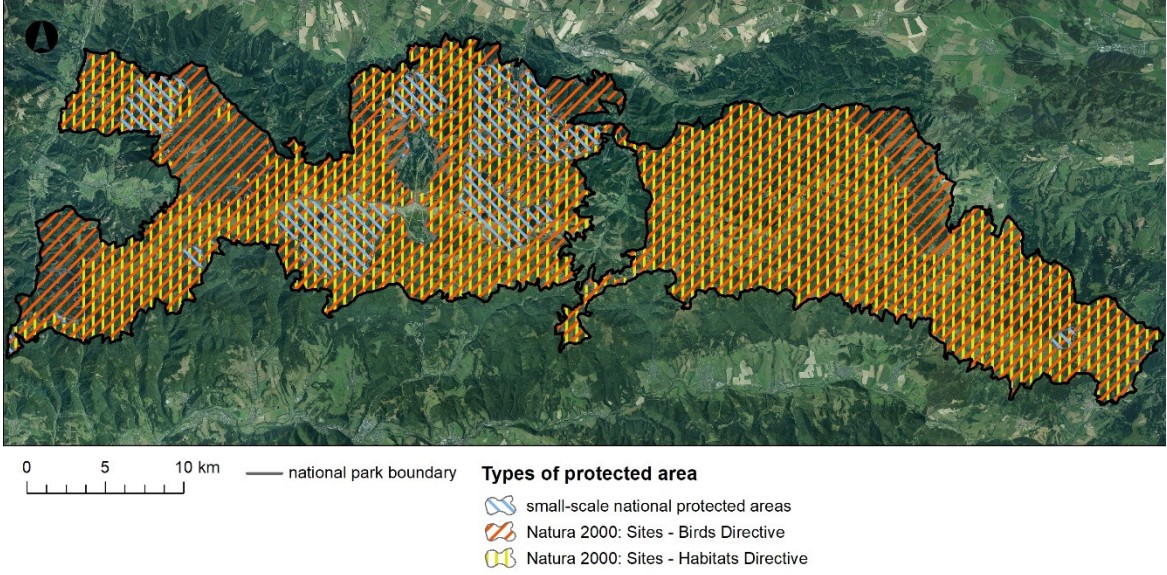

**Figure 2.** Protected areas of Low Tatras National Park.

National park boundaries were changed mainly due to property-law relations 19 years later, in 1997. The area of the national park was 72,842 ha, and the area of the protection zone was 110,162 ha. Approximately 11,000 ha of the most valuable parts of its territory are strictly protected (included within the 4th and the 5th zone of protection-A and B zone).

Despite the lengthy and complicated process, the declaration of the national park had been a successful culmination of many years of effort of a large number of professional and voluntary nature conservationists, various experts, as well as simple supporters of this unique mountain range in the middle of Slovakia.

### 2.2. Data

For a long time, aerial and satellite imagery has been used to detect and classify landscape transformations over time, as it is useful to capture the impacts of many processes causing natural (e.g., fires, wind disasters) and anthropogenic (e.g., deforestation, urbanization, agriculture) changes [40]. The CORINE land cover (CLC) data for 1990 and 2018, created by visual interpretation of high-resolution satellite images, were applied in our research. CLC data uses a minimum mapping unit (MMU) of 25 hectares (ha) for areal phenomena and a minimum width of 100 m for linear phenomena. The CLC change layers, which highlight changes in land cover with an MMU of 5 ha, have been used to monitor land cover changes. Different MMUs mean that the change layer has a higher resolution than the status layer (Table 1) [41].

**Table 1.** The specifications of the CORINE land cover data in the reference years 1990 and 2018.

| Specification | CLC 1990 | CLC 2018 |
|---|---|---|
| Geometric accuracy, satellite data | ≤50 m | ≤10 m |
| Min. mapping unit/width | 25 ha/100 m | 25 ha/100 m |
| Geometric accuracy, CLC | 100 m | Better than 100 m |
| Thematic accuracy, CLC | ≥85% | ≥85% |

Using this data, we were able to capture and analyze changes occurring in almost 30 years. We have identified 13 land cover classes within the studied area (Table 2). We have used RGB color codes defined by EEA in the maps of land cover.

**Table 2.** CORINE land cover classes in the studied area.

| Level 1 | Level 2 | Level 3 |
|---|---|---|
| 1 Artificial surfaces | 11 Urban fabric | 112 Discontinuous urban fabric |
| | 14 Artificial, non-agricultural vegetated areas | 142 Sport and leisure facilities |
| 2 Agricultural areas | 21 Arable land | 211 Non-irrigated arable land |
| | 23 Pastures | 231 Pastures |
| | 24 Heterogeneous agricultural areas | 243 Land principally occupied by agriculture, with significant areas of natural vegetation |
| 3 Forest and semi-natural areas | 31 Forests | 311 Broad-leaved forest |
| | | 312 Coniferous forest |
| | | 313 Mixed forest |
| | 32 Scrub and/or herbaceous vegetation associations | 321 Natural grasslands |
| | | 322 Moors and heathland |
| | | 324 Transitional woodland-shrub |
| | 33 Open spaces with little or no vegetation | 333 Sparsely vegetated areas |
| 5 Water bodies | 51 Inland waters | 512 Water bodies |

*2.3. Methods*

We have used the ArcMap 10.5 software in our research, in which the land cover maps from 1990 and 2018 and maps of land cover changes using available CLC change layers had been made. We have focused mainly on the percentual evaluation of changes in the individual elements of the land cover and analyzed them statistically and spatially. We have made a cross-table to express qualitative relationships between the two variables [42,43]. Using this method, we have found out which land cover classes had been changed and to which classes they had been modified at the same time. Thus, we were able to identify the core processes that took place in the landscape. At the same time, we can identify the period in which the most changes took place and which left the most significant consequences on the landscape.

Landscape changes were categorized into so-called "land cover flows (LCFs)", i.e., classes that reflect processes taking place in the observed area. The definition of these changes was studied by many scientists, who have defined different amounts and types of land cover flows [44–47].

The most extensive and detailed categorization was introduced by Haines-Young and Weber in 2006 [48], defining nine types of changes:

- LCF1 Urban land management—internal transformation of urban areas
- LCF2 Urban residential sprawl—land uptake by residential buildings altogether with associated services and urban infrastructure (classified in CLC111 and 112) from non-urban land (extension over sea may happen)
- LCF3 Sprawl of economic sites and infrastructures—sprawl of economic sites and infrastructures: Land uptake by new economic sites and infrastructures (including sport and leisure facilities) from non-urban land (extension over sea may happen)
- LCF4 Agriculture internal conversions—conversion between farming types. Rotation between annual crops is not monitored by CLC
- LCF5 Conversion from forested and natural land to agriculture—extension of agriculture land use
- LCF6 Withdrawal of farming—farmland abandonment and other conversions from agriculture activity in favor of forests or natural land

- LCF7 Forests creation and management—creation of forests and management of the forest territory by felling and replanting. Due to the CLC cycle of 10 years, only one part of the shrubs is tall enough to be identified as trees. In order to take stock of all recent plantations, conversions of semi-natural land to CLC324 are conventionally recorded as afforestation (although some natural colonization may take place)
- LCF8 Water bodies creation and management—creation of dams and reservoirs and possible consequences of the management of the water resource on the water surface area
- LCF9 Changes of land cover due to natural and multiple causes—changes in land cover resulting from natural phenomena with or without any human influence.

This categorization was chosen for its detail and complexity for our research. Dominant processes in the landscape of the Low Tatras National Park can be identified based on the percentual data of individual changes. Land cover flows summarize and interpret all possible one-to-one changes between the CORINE land cover classes. The changes are grouped into so-called flows and are classified according to major land-use processes. We have focused on the main class of land cover flows (e.g., LCF7), which consist of several subclasses of land cover flows (LCF71, LCF72, LCF73 and LCF74) (Figure 3).

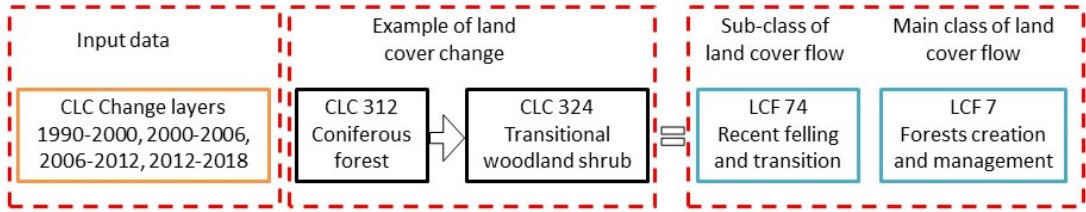

**Figure 3.** Methodology of the land cover flows.

The last part of our research consisted of assessing landscape structure changes based on landscape metrics calculations using the PatchAnalyst tool. Patch Analyst is an ArcGIS extension that facilitates spatial analysis of landscape patches. It is used for spatial pattern analysis, often in support of habitat modeling, biodiversity conservation, and forest management. The software offers analyses of several types of landscape-ecological metrics, which are often used in scientific research, primarily to assess landscape fragmentation in Slovakia [49–51] and abroad [52–54]. In addition to national or regional spatial data [55], CORINE land cover data, which are also applicable to regional research, are suitable and frequent input data for the calculation of landscape metrics. However, they are not suitable for research of a smaller area due to their lower accuracy. Krajewski [56] distinguishes three approaches to the study of landscape changes: identification of spatiotemporal changes [57], identification of driving forces of changes [2], and identification of landscape changes based on landscape metrics [52,58]. In our research, we combine all three approaches and make a comprehensive analysis of changes in land cover in the Low Tatras National Park. Following Kumar et al. [52], Obeidat et al. [59], Singh et al. [53], we have not used all of the indices from the PatchAnalyst tool because some of them are closely related and some are redundant. The selection was made following the intention of the study based on the knowledge of previous research. An important principle is to select unrelated indices. The following six indices were selected:

- Number of patches (NumP)—a simple indicator that indicates the total increase, respectively, a decrease in the number of patches in all categories in the observed area.
- Mean patch size (MPS)—average patch size. This indicator shows the disintegration of the spatial structure of the landscape.
- Total edge (TE)—an indicator that represents the sum of perimeters of all patches.
- Area weighted mean shape index (AWMSI)—an index that reflects the shape complexity of patches. The index is equal to 1 if the patches have a circular or square shape. The index value increases if the shape is irregular. It differs from the "mean shape index" metric by assigning different weights to individual patches (the larger the area, the higher the weight).

- Shannon's diversity index (SDI)—an index that determines landscape diversity calculated as the proportions of the land cover classes across the total area. SDI increases by the number of patches in the landscape feature categories. The higher the index value, the higher the landscape heterogeneity, i.e., the landscape is more abundant in the number of categories of landscape features and the number of patches [60]. The index will be equal to 0 when there is only one patch in the landscape and increases as the number of patch types or the proportional distribution of patch types increases.
- Shannon's evenness index (SEI)—an index that determines the distribution of patches and their abundance. A proportional reduction in the number of patches and categories also causes a reduction in the overall balance. The landscape metrics balance within the observed landscape is better when the value of this index converges to one.

It is important to evaluate changes in the landscape, especially in national parks, from the ecological point of view, too. The interpretation of the quantified data is important to determine ecological signification and the current state of the landscape, e.g., [52,53,56,58,59].

## 3. Results

### 3.1. Land Cover of Low Tatras National Park in 1990 and 2018

There were 13 land cover classes in 1990 and one less in 2018 in the observed area. The land cover of the Low Tatras National Park in 1990 and 2018 is shown in Figure 4.

Almost two-thirds of the National Park area was covered with coniferous forests (CLC 312) in 1990, which consisted of various closed formations of conifers-spruce, fir, pine, and larch. A sample of the class was formed by islands of trees of the abovementioned species, alternating individually or in groups, represented by several types of conifers. They were complemented by fragments of incidental deciduous trees, grasslands of forest meadows, shrubs, transitional woodland-shrubs, forest roads, or parts of recreational facilities, scattered settlements, croplands [61] and abandoned mining sites. Approximately 11% of the National Park area was covered by natural grasslands (CLC 321), consisting mainly of alpine meadows. These areas were not primarily suited for agricultural function, and their natural development was not inhibited by human influence. Meadows were sometimes supplemented by dwarf mountain pine growths, rocks, or groups of trees and shrubs [60]. Nearly 9% of the area was occupied by mixed forests (CLC 313), consisting mainly of spruce, pine, larch, beech, oak, maple, birch, and other tree species. The sample of mixed forests consisted of groups of alternating solitary individuals or islands of coniferous and deciduous trees [61]. Approximately 6% of the area was covered by dwarf mountain pine (CLC 322) sporadically interrupted by enclaves of rock relief forms and alpine meadows. Its occurrence was related to the top parts of the Low Tatras mountain ridge [61].

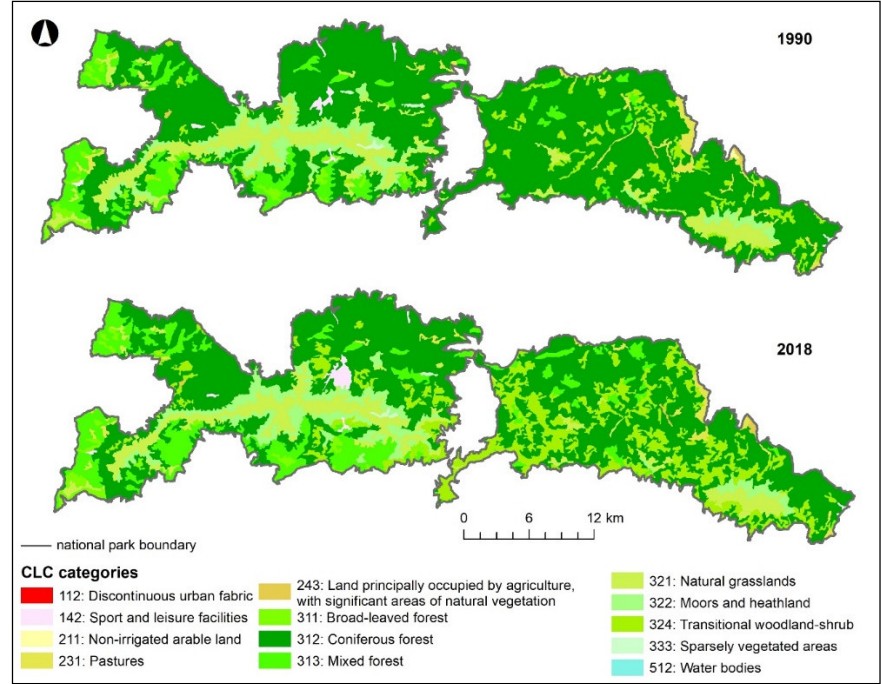

**Figure 4.** Land cover of Low Tatras National Park in 1990 and 2018.

Almost 5% of the area was covered with transitional woodland-shrub. Young forest trees (deciduous and coniferous) planted after loggings or various calamities were mainly represented in this class, together with forest nurseries, naturally developed forest formations (shrubs and herbaceous vegetation with scattered trees), or shrub formations on abandoned meadows, pastures and forest cuttings for high-voltage power lines. A representative sample of the class consisted of alternating coppice belts and forest remnants (the areal representation of coppice within the respective patches reached 60% or more) [61]. Other land cover classes had not reached values higher than 3% (Table 3).

**Table 3.** Land cover classes of Low Tatras National Park in 1990 and 2018.

| CLC | 1990 | | 2018 | |
|---|---|---|---|---|
| | Area in ha | % | Area in ha | % |
| 112 | 6.39 | 0.01 | 0.46 | 0.00 |
| 142 | 421.92 | 0.56 | 576.05 | 0.76 |
| 211 | 47.48 | 0.06 | - | 0.00 |
| 231 | 1641.91 | 2.16 | 1004.76 | 1.32 |
| 243 | 203.73 | 0.27 | 228.45 | 0.30 |
| 311 | 1458.41 | 1.92 | 1286.17 | 1.69 |
| 312 | 48,913.12 | 64.35 | 39,958.69 | 52.57 |
| 313 | 6814.73 | 8.96 | 9114.03 | 12.00 |
| 321 | 8392.56 | 11.04 | 6632.16 | 8.72 |
| 322 | 4330.54 | 5.70 | 5062.95 | 6.66 |
| 324 | 3597.34 | 4.73 | 12,021.51 | 15.81 |
| 333 | 185.37 | 0.24 | 131.21 | 0.17 |
| 512 | 3.05 | 0.00 | 0.11 | 0.00 |
| Total | 76,016.55 | 100.00 | 76,016.55 | 100.00 |

Legend: 112 Discontinuous urban fabric, 142 Sport and leisure facilities, 211 Non-irrigated arable land, 231 Pastures, 243 Land principally occupied by agriculture, with significant areas of natural vegetation, 311 Broad-leaved forest, 312 Coniferous forest, 313 Mixed forest, 321 Natural grasslands, 322 Moors and heathland, 324 Transitional woodland-shrub, 333 Sparsely vegetated areas, 512 Water bodies.

*3.2. Land Cover Flows in Low Tatras National Park between 1990 and 2018*

Five main processes were identified in the studied area between 1990 and 2018. LCF7 had the largest share in terms of the proportion of the total area changed (Table 4, Figure 5). This process was clearly dominant in all monitored periods and is mainly represented by two processes: wind calamities and grazing.

An increase of CLC class 324 (transitional woodland-shrub) on one hand, and a significant decline of CLC 312 (coniferous forests) on the other, was observed due to a frequent occurrence of wind calamities in the recent years, which are a result of widespread climate change, not only on a global but also on a local scale in the last three decades. Mountain ranges of the Carpathian Arch are no exception. Recurring extreme climatic situations, which occur in the observed area, are becoming increasingly frequent [62], and the forest stands are destructively affected mostly by windstorms [63].

Several massive windstorms have swept through the mountain ridge of the Low Tatras belonging to the Low Tatras National Park over the past 25 years, causing vast windfalls in the spruce monocultures forest growths, especially in the eastern part (Kráľovohoľská part) of the mountain range (National Park). Since the beginning of the studied period, windstorms with an impact on the spread of spruce monocultures had been recorded on 8 July 1996 (wind calamity Ivan), 27–28 October 2002 (wind calamity Sabina), 16–17 November 2002 (wind calamity Klaudia), 19 November 2004 (wind calamity Alžbeta), 18–19 January 2007 (wind calamity Kyrill), 23–24 August 2007 (wind calamity Filip), 17–19 May 2010 (wind calamity Gizela) [63–65]. The last more extensive one occurred on 14–15 May 2014 (windstorm Žofia) [66].

It is logical to conclude that the consequences of windstorms manifested by large-scale windfalls will be reduced significantly in the future years, since the critical relief sites overgrown with monocultural spruce forests, which were exposed to impact air currents, have been replaced mainly by transitional woodland-shrub.

Relatively extensive changes in vegetation have also been recorded in the zone of the (anthropogenically created) timberline in the subalpine level, in addition to the extensive area changes caused by windstorms scattered throughout the National Park (mountain range). The dwarf mountain pine belt (CLC 312) experienced an area increase of almost 1%. The phenomenon of windstorms was also marginally present here, causing a slight retreat of monocultural coniferous forests which were replaced by transitional woodland-shrub (CLC 324) on the timberline, especially in the area of Veľký Gápeľ and Malý Gápeľ, as well as in the area of the northern and southern slopes of Lajštroch.

**Table 4.** Land cover flows (LCF) in the Low Tatras National Park between 1990 and 2018.

| LCF | | 1990–2000 | 2000–2006 | 2006–2012 | 2012–2018 | 1990–2018 | CLC Classes Changes |
|---|---|---|---|---|---|---|---|
| LCF3 | ha | 0.00 | 0.00 | 13.80 | 193.32 | 207.12 | 312–142 |
| | % | 0.00% | 0.00% | 0.19% | 6.50% | 1.38% | |
| LCF4 | ha | 0.00 | 54.01 | 0.00 | 0.00 | 54.01 | 211–231 |
| | % | 0.00% | 1.80% | 0.00% | 0.00% | 0.36% | |
| LCF5 | ha | 9.07 | 0.00 | 0.00 | 0.00 | 9.07 | 243–211 |
| | % | 0.48% | 0.00% | 0.00% | 0.00% | 0.06% | |
| LCF6 | ha | 133.98 | 66.00 | 0.00 | 0.00 | 199.98 | 231–324 |
| | % | 7.14% | 2.20% | 0.00% | 0.00% | 1.33% | |
| LCF7 | ha | 1732.54 | 2901.05 | 7170.29 | 2778.76 | 14,582.64 | 311–324, 312–324, 313–324, 313–311, 321–324, 324–312, 324–313, 324–311 |
| | % | 92.37% | 96.03% | 99.81% | 93.50% | 96.88% | |
| change of total area (%) | | 2.47% | 3.97% | 9.45% | 3.91% | 19.80% | - |

Land cover flows in the anthropogenically lowered (current) timberline had been affected by grazing of sheep and cattle in the past decades (before the beginning of the studied period). Currently, climate change is seen as a significant phenomenon of dwarf mountain pine expansion

and acceleration of succession in the subalpine level of mountain meadows, as well as the shifting of the timberline to its original altitude.

Relatively significant changes in the area extent of individual types of land cover have occurred on the timberline in the high mountain ranges of the Western Carpathians, including the Low Tatras, in recent decades. According to R. Midriak's research, the tree and dwarf mountain pine vegetation at the timberline expanded by up to 6% at the expense of the subalpine mountain meadows level in the decade from 1990 to 2000 [67]. These changes occurred due to the end of sheep and cattle grazing above the timberline. Attenuation of grazing began during the 60 s and 70 s of the 20th century, especially in the Kráľovohoľská (eastern) part of the Low Tatras. Decrease of grazing is the reason why the succession at the timberline and the shift of the anthropogenic timberline to higher altitudes are more pronounced in the eastern than in the western (Ďumbierska) part of the mountain range [68], where grazing finally disappeared only after 1989.

Up to the 14th century, the original-natural timberline, which existed before anthropogenic interventions of shepherds, ascended to an altitude of 1600 m a.s.l., even up to 1700 m a.s.l. in some parts of the mountain range [69–71]. Artificial timberline in the Low Tatras oscillates nowadays between altitudes of 1300–1520 m a.s.l. [67,70]. Climate change also plays a role in the land cover development and thus significantly shifts the timberline to higher altitudes in addition to the end of grazing and the natural course of succession. These processes create much more favorable ecological conditions for the growth of dwarf mountain pine or spruce trees on the timberline. For this reason, we expect an increase in dwarf mountain pine formations, as well as the expansion of transitional woodland-shrub with the potential of their conversion to coniferous (spruce) forest formations to higher altitudes in the coming years or decades [72,73]. The average temperature in mountain areas of the Western Carpathians should increase by at least 2 °C over 30 to 50 years, according to climatological models [74]. Longer periods of droughts should also occur. Under such conditions, the montane zones (800–1200 m) will be unsuitable for the natural occurrence of spruce, which will shift to higher altitudes.

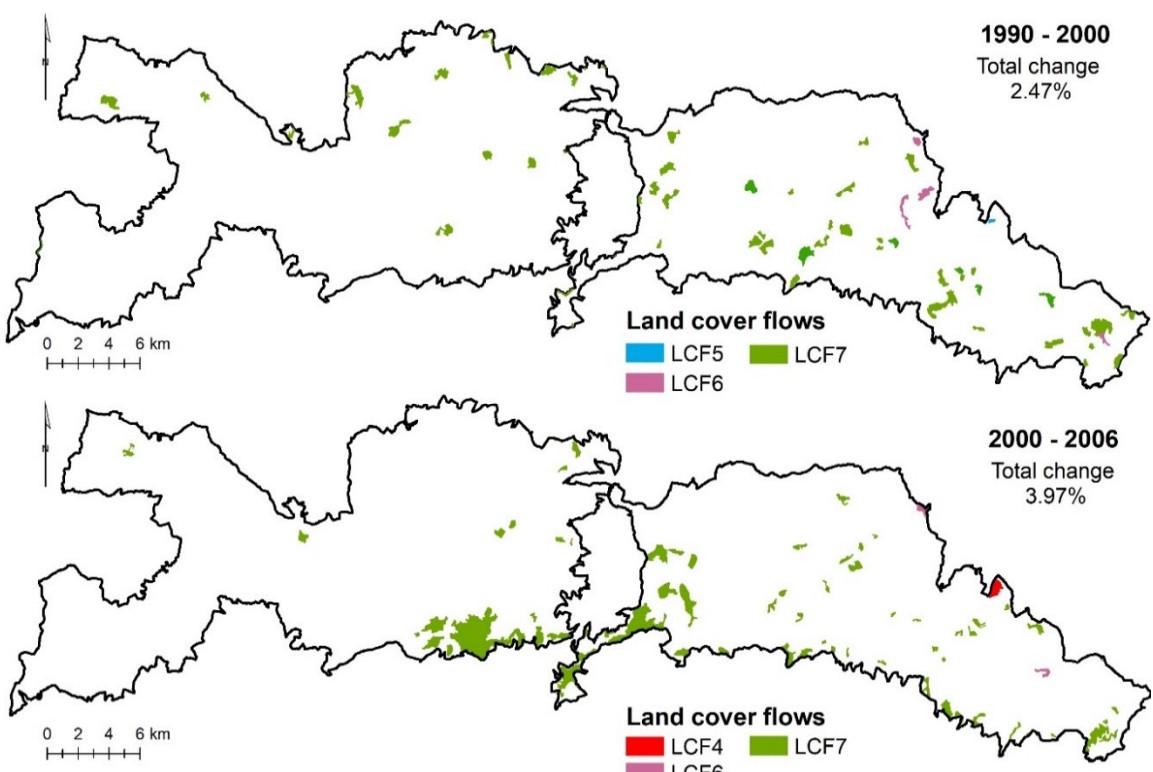

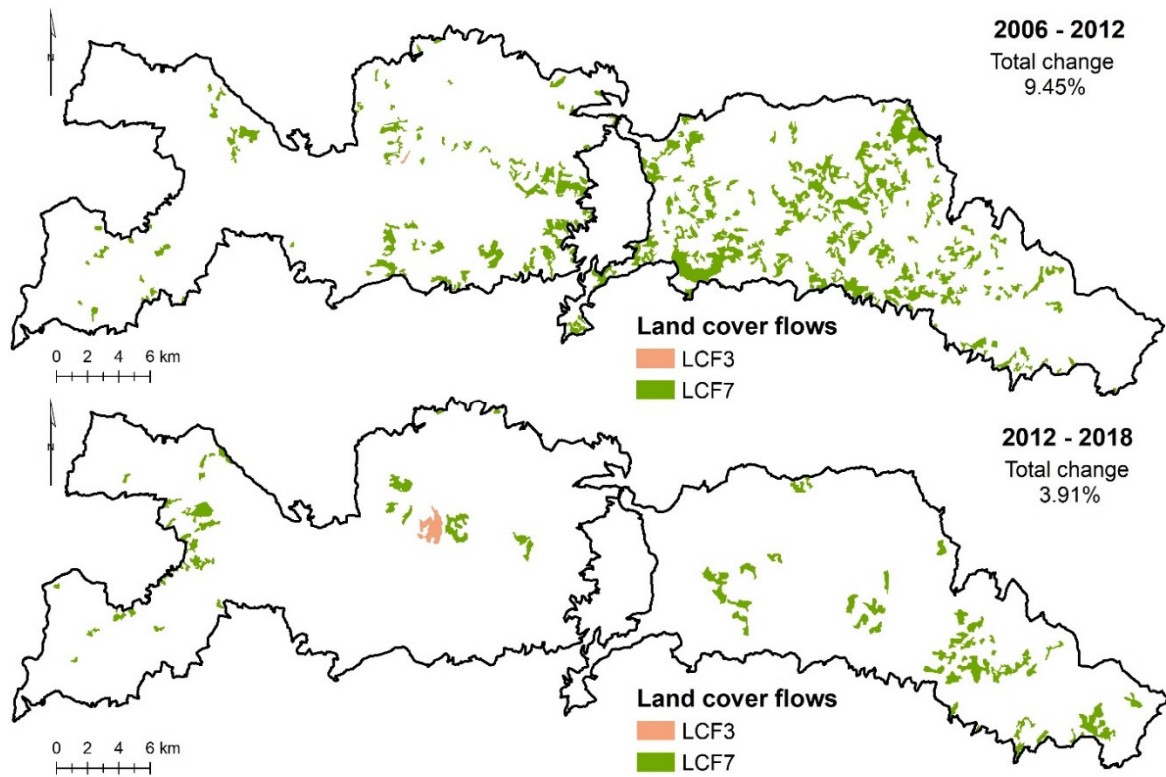

**Figure 5.** Land cover flows in Low Tatras National Park between 1990, 2000, 2006, 2012 and 2018.

The increase of the mixed forest (CLC class 313 mixed forest) during the observed period is linked to montane and foothill areas, which were used as agricultural areas with 80% permanent grassland in the second half of the 20th century [75].

Significant changes in the long-term use of pastures in this landscape area occurred after 1989. The transformation of the political-economic situation and the ownership relations has led to a rapid reduction in the numbers of cattle grazing to almost zero and a gradual reduction of sheep herds, which are currently grazed here only rarely. This trend has caused the beginning of a rapid succession downwards into the landscape parts, from which the mixed or deciduous forests have been pushed out as a result of extensive agriculture in the recent centuries [76].

These rapid changes in the long-term use of permanent grassland-agricultural land after 1989 triggered the abandonment of foothills and mountain pastures, which caused a rapid spontaneous overgrowth of the landscape (succession). The result can be seen as a 3% increase in the CLC class 313 mixed forest after less than thirty years. All of these factors are understood as driving forces within the DPSIR model developed by the European Environment Agency [77]. They represent the triggering mechanisms of landscape processes, which induce area extent changes of each CLC category.

### 3.3. Assessment of Land Cover Changes Based on Landscape Metrics

Landscape metrics is the most commonly used tool to compare the evolution of land cover changes over time. The values of the selected landscape indices calculated at the landscape level are shown in Table 5.

**Table 5.** Landscape metrics changes between 1990 and 2018 at the landscape level.

| Landscape Metric | 1990 | 2018 | Change | Manifestation in Landscape |
|---|---|---|---|---|
| NumP | 275 | 281 | +6 | a small increase in the number of patches |
| MPS | 276.536 | 270.521 | −6.015 | a more fragmented landscape |
| TE | 2,998,750 | 3,609,590 | +610,840 | an increase in the total perimeter of patches |
| AWMSI | 9.65107 | 6.16634 | −3.48473 | a decrease in the heterogeneity of patch shapes and a higher regularity of the shapes of larger patches |
| SDI | 1.27467 | 1.46898 | +0.19431 | an increase in landscape heterogeneity |
| SEI | 0.496958 | 0.591159 | +0.094201 | an increase in landscape balance |

A small increase in the number of patches (NumP) was observed, indicating only a slight increase in the heterogeneity of the landscape. The MPS index values point to a higher degree of landscape fragmentation in 2018. This finding is also reflected in the value of the TE index, which has shown a visible increase. The AWMSI index indicates a decrease in the heterogeneity of patch shapes, which may be caused by regular rectangular or oval shapes of patches created in the process of salvage collection. The Shannon Diversity Index was higher in 2018 than in 1990. However, only a minimal increase can be seen, which points to a higher balance in the proportions of the landscape features. The Shannon Equilibrium Index, which is complementary to the SDI, increased by almost 0.1 over the studied period. Thus, we can speak of a slight trend in an increasing uniformity of patch expansion in the landscape mosaic classes.

## 4. Discussion

The main research questions asked in the introduction of this study can be answered based on analyses made during the research.

As already mentioned and presented in Table 4, five significant land cover flows have been identified in the Low Tatras National Park between 1990 and 2018. The most significant land cover flow (96.88%) was the LCF 7 Forest creation and management, in which up to eight types of processes took place (Table 4). Area changes of all classes are spatially located throughout the national park, but to a greater extent in the Kráľovohoľská part (Figure 5). The main reason for their predominant location in the eastern part of the National Park lies in its geographical location. Land cover in the form of monocultural forests is the result of intensive forestry during the second half of the 20th century. Spruce monocultures on the exposed mountain slopes cannot withstand the wind as the original anthropogenically removed forests [32,62,67,73].

The massive windstorms in the prevailing north-western streaming naturally oscillated on the northern slopes of the mountain range, after running off the Tatra's massive vault, to impact the large-scale artificial monocultural spruce formations located at inappropriate relief sites with catastrophic consequences. This process was dominant in all four monitored periods, but mostly in 2006–2012, which is the result of frequent wind disasters.

LCF 6 Withdrawal of farming (1.33%) was shown in the CLC class 231 pastures (meadows and pastures), which was transformed into CLC 324 (transitional woodland-shrub). The end of the almost 500 years of farm animals grazing was very rapid at the end of the 20th century. The grazing ban was related to legal regulation in connection with the protection of the landscape within the national park. Farm animals have disappeared from alpine pastures, which have been subject to intense succession since then [67,68,76,78–80].

This process is closely related and complementary to the previous land cover flow. It is reflected in a greater extent again in the Kráľovohoľská part of the national park, significantly in the entire valley of Ipoltica, in the broader hinterland of Liptovská Teplička, in the vicinity of the Čierny Váh water reservoir and the transformed areas to the east extend up to the main ridge near Kráľova Hoľa. This process occurred only in the period 1990 to 2006. In addition to wind disasters, the reason is also the human factor (logging).

LCF 5 Conversion from forested and natural land to agriculture is the inverse process to the previous one, although it is of lower intensity (0.06%). CLC class 243 land principally occupied by agriculture, with significant areas of natural vegetation, has been transformed into CLC class 211 non-irrigated arable land. This process was identified only in the period 1990–2000 in the vicinity of the village Liptovská Teplička as a conversion from agriculture–nature mosaics to continuous agriculture [81,82]. The cause of this transformation must be sought within links of ownership in the second half of the 20th century. After the transformation of the original strip fields into large agricultural areas used as meadows and pastures, these were abandoned after 1989. The original owners have gradually begun to use them again during the period studied. Original meadows and pastures in higher altitudes succumbed to succession and have been transformed into forests, while the original fields of arable land in the lower altitudes have been transformed and are currently used as grasslands–meadows, and pastures [83].

A surprising result of our research was the finding that the development of touristic centers with high demands on recreational infra and suprastructure does not manifest itself as a significant land cover flow. LCF 3 Sprawl of economic sites and infrastructures has transformed the CLC class 312 coniferous forests into the CLC class 142 sport and leisure facilities, but this impact is not significant compared to changes caused by wind calamities or agricultural land use. The most significant changes have been identified on the southern and northern slopes of Chopok [84–86], where the ski resort Jasná was built. Tatry mountain resorts, Inc. operates 23 lifts and 39 ski slopes in the largest resort in Slovakia. Other location of transformations can be found in the territory of Demänovská Valley (Demänovská Ice Cave).

LCF 4 Agriculture internal conversions (0.36%) was recorded only in the period 2000–2006 in the vicinity of the village Liptovská Teplička. Arable land was transformed into meadows and pastures in this locality. Based on the landscape-ecological metrics results, the studied area showed an increase in the land cover heterogeneity, although the shape of patches was more regular. These results also showed that the landscape of the national park has significantly lost its forest potential at the expense of less valuable forest formations over the studied period. Analyses have shown that the timberline shifted to higher altitudes, and there is a trend of a continual succession of alpine grasslands. Although most of the landscape metrics results were positive, it is not possible to draw more profound conclusions from them. According to Ružičková et al. [78], the resulting values of the landscape diversity index do not describe the ecological stability and quality of the assessed area and do not take into account the internal differentiation of landscape structure features. For this reason, we do not refer to our calculation results as absolute, and we consider the different (and changing) quality and structure of landscape features over time as well. We can expect extensive linear and areal interventions into the current land cover based on the expected future changes that will occur due to the planned construction of new transportation projects and the technical infrastructure connected to tourism (ski cableways in Demänovská Dolina resort and sports hall in Donovaly resort). Combined with the anthropogenically predisposed development of the timberline and the progress of succession, the CLC class 142 Sport and leisure facilities will increase at the expense of classes 312 Coniferous forests, 324 Transitional woodland-shrub, and 333 Sparsely vegetated areas in the critical construction localities. The construction of express road R1 section Slovenská Ľupča-Liptovská Osada will affect CLC classes at lower altitudes [87]. A decrease in the areal spread of the classes 231 Pastures, 311 Broad-leaved forest, and 313 Mixed forests is predicted.

Landscape metrics were used in this research because they have been providing a backbone for spatial pattern analysis in landscape ecology for more than three decades. It is very important to select the correct approach, or combination of approaches, for investigating the issue [88]. On the basis of landscape metrics results, we could contend that area of the National Park is more heterogenous, uniform and balanced. These conclusions are very one-sided; therefore, it is required asses the landscape changes according to several approaches. Changes in the landscape of the national park caused by radical interventions (natural and anthropogenic) were reflected in an increase of diversity, but they may also have an impact on ecological stability. These values should be interpreted sensitively because it does not take the internal differentiation of land cover classes

into account. Assessment of land cover changes is especially relevant for protected areas where long-term ecosystem stability is a critical aspect of protecting and maintaining high levels of biodiversity and ecosystem functions [89].

Based on the results of our research, we can formulate basic recommendations for the management of the National Park concerning the negative processes caused by unwanted changes in the land cover resulting from the results and conclusions of the study. The priorities should be:

- To stop the deterioration of habitat status, in particular, for the habitats of European and national importance, maintain their current state, and then take steps towards a measurable improvement. Therefore, the National Park Administration should give priority to the detailed mapping of the habitat status, complete an overall map of the National Park and provide operational data for decision-making by state administration authorities, in a particular state and private forest managers;
- To map in detail the natural forests and primeval forests relics of the National Park in the shortest possible time to ensure that their area extent is maintained and to gradually increase their extent of areas with a potential for natural forest development;
- Prevent further fragmentation of forests and encourage their regeneration while ensuring compensatory mechanisms to cover the loss of forest management and favoring alternative uses of high nature value forests;
- Implement measures to preserve and improve habitats of European importance, particularly in Natura 2000 sites and habitats of national importance within the National Park;
- Improve the effectiveness of communication between the environmental and agricultural departments;
- Define or revise the nature and landscape conservation objectives in the National Park in more detail.

## 5. Conclusions

Based on the analyses of changes in land cover transformation over the observed period, we can conclude that wind calamities were the main transformation factor of national park landcover changes between 1990 and 2018. Their destructive power stems from improper forest management in the second half of the 20th century in combination with anthropogenic climate change. The ending of livestock grazing on foothills, but also montane pastures, was also an essential factor. The end of grazing triggered succession towards lower, as well as higher altitudes, to the original forest habitats. At the end of the 1980s, this disrupted the landscape balance of the National Park, maintained by humans since the Middle Ages. Changes in agricultural management at the foothills of the National Park were another essential impulse in the transformation of land cover. Last but not least, the development of tourism and the growth of recreational infrastructure have been among the most intensive transforming factors of land cover changes in recent decades.

Analyses of land cover changes over the last 30 years in the Low Tatras National Park have clearly pointed to the inaccuracy of forest management and planting of monocultural forests, especially on the northern slopes, which are most exposed to extreme wind situations. The combination of these two factors has the most negative effect on the alpine country. When managing the forests of a national park, emphasis must be placed on the species composition of forests, which should be as close as possible to the composition of the original forests. After wind calamities, monocultural spruces should be replaced by beech and beech-fir forests.

A plan for making the zones of the national park in terms of its economic use should also be drawn up. Currently, the Ministry of the Environment of the Slovak Republic is working intensively on it. The proper delimitation of individual zones of use, especially in relation to tourism, could significantly prevent the expansion of recreational areas at the expense of the surrounding countryside, where the main example is the Demämovská valley on the northern side of the mountain range.

Our conclusions clearly show that analyses of land cover and land cover flows can contribute to the proper planning of land use in national parks and thus to its stabilization and sustainability.

**Author Contributions:** Conceptualization, M.Ž.; methodology, M.Ž., B.G. and P.H.; software, M.Ž.; validation, M.Ž., B.G. and P.H.; formal analysis, M.Ž. ; investigation, B.G. and P.H.; resources, M.Ž., B.G. and P.H.; data curation, M.Ž.; writing—original draft preparation, M.Ž., B.G. and P.H.; writing—review and editing, M.Ž., B.G. and P.H.; visualization, M.Ž.; supervision, M.Ž.; project administration, B.G.; funding acquisition, B.G. All authors have read and agreed to the published version of the manuscript.

**Funding:** This research and the APC were funded by Slovak Research and Development Agency, grant number APVV-18-0185 "Land-use changes of Slovak cultural landscape and prediction of its further development" and by VEGA, grant number 1/0236/18 "Environmental aspects of mining localities settings in Slovakia in Middle Ages and the beginning of Modern history".

**Conflicts of Interest:** The authors declare no conflicts of interest. The funders had no role in the design of the study; in the collection, analyses, or interpretation of data; in the writing of the manuscript, or in the decision to publish the results.

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
