# Peer review of "Mapping of the Land Cover Changes in High Mountains of Western Carpathians between 1990–2018: Case Study of the Low Tatras National Park (Slovakia)"

_land, doi:10.3390/land9120483_

Round 1
Reviewer 1 Report
The article MAPPING OF THE LAND COVER CHANGES IN HIGH MOUNTAINS OF WESTERN CARPATHIANS BETWEEN 1990 - 2018: CASE STUDY OF THE LOW TATRAS NATIONAL PARK (SLOVAKIA) presents a topical issue, which aims to analyze the landscape changes in the study area and to propose solutions for long-term sustainable development. Changes were monitored using three methods and the proposed objectives were achieved.
In order to be published I recommend a few additions:
In row 20 in the abstract, correct Corin with Corine
In paragraph 2.2 where the characteristics of the CLC database are presented, the source of the information is not specified. The technical guide prepared by the European Environment Agency should also be included in the bibliography.
In the conclusions part, among the recommendations made to the National Park Administration, mention the Natura 2000 sites. In order to have an overview of these conservation and protection areas, I would recommend their representation on maps.
Best regards!
Author Response
Rewiever 1
The article MAPPING OF THE LAND COVER CHANGES IN HIGH MOUNTAINS OF WESTERN CARPATHIANS BETWEEN 1990 - 2018: CASE STUDY OF THE LOW TATRAS NATIONAL PARK (SLOVAKIA) presents a topical issue, which aims to analyze the landscape changes in the study area and to propose solutions for long-term sustainable development. Changes were monitored using three methods and the proposed objectives were achieved.
In order to be published I recommend a few additions:
In row 20 in the abstract, correct Corin with Corine → revised
In paragraph 2.2 where the characteristics of the CLC database are presented, the source of the information is not specified. The technical guide prepared by the European Environment Agency should also be included in the bibliography. → reference of Technical Guidelines of CLC was added to the text and to the Bibliography. It was the source of main characteristics of the CLC mentioned in paragraph 2.2.
Büttner, G.; Kosztra, B.; Soukup, T.; Sousa, A.; Langanke, T. CLC2018 Technical Guidelines; EEA: Copenhagen, Denmark, 2017. Available online: https://land.copernicus.eu/user-corner/technical-library/clc2018technicalguidelines_final.pdf (accessed 20 October 2020)
In the conclusions part, among the recommendations made to the National Park Administration, mention the Natura 2000 sites. In order to have an overview of these conservation and protection areas, I would recommend their representation on maps. → Map of protected areas of the Low Tatras National Park (national small-scale protected areas and areas of NATURA 2000) was added to the chapter 2.1.

Reviewer 2 Report
I think that this paper makes a valuable contribution to environmental monitoring in Slovakia, but substantial grammatical revisions are necessary to prepare it for publication. My specific suggestions are outlined below.
Abstract:
Page 1, Line 20: I would recommend adding "in the Slovak Republic" after "Low Tatras National Park", as many readers may not know where the Low Tatras National Park is located.
Page 1, Line 20: I would recommend removing the word "the" before "Corin Land Cover".
Page 1, Line 20: I think that "Corin Land Cover" should be changed to "Corine Land Cover".
Introduction:
Page 1, Line 37: I would recommend removing the comma between "Asia" and "South America" and adding the word "and".
Page 1, Line 38: I would recommend removing the "as well" after "Slovakia".
Page 1, Line 40: I would recommend changing "their leading causes" to "the leading causes of land cover changes".
Page 1, Line 42: I would recommend changing "had been" to "was".
Page 2, Line 53-56: This is more a matter of personal taste, so please feel free to disregard: I found the "On the one hand ... on the other hand" structure of this sentence confusing. I would recommend changing the sentence to something more like the following: "One prominent body of research on the mountain environment in Slovakia focuses on geodynamic and geomorphological processes. A second prominent body of research focuses on changes in the landscape structure..."
Page 2, Line 62: Again, this is more a matter of personal taste, but I think the phrase "significantly essential" is redundant. I would recommend removing "significantly".
Page 2, Line 72: The CLC is considered to be the "most complex" what? Please specify.
Page 2, Line 76: I think that "valuable" may not be the most appropriate word choice here. Would "accurate" or "complete" or "sophisticated" be more appropriate to the author's meaning?
Page 2, Line 83-84: How did the authors decide upon these time periods? Was it based on data availability or historical context?
Materials and Methods:
Page 3, Line 109: I would recommend specifying in the caption of Figure 1 that the highlighted region in the figure is the Slovak Republic.
Page 3, Line 119 - Page 5, Line 177: Since the focus of this paper is on land cover changes, I don't think so much detail about the geology, climate, rainfall, etc of the park is necessary. I would recommending revising this section to make it shorter and more streamlined.
Page 6, Line 246 - Page 7, Line 282: I found it a little bit confusing that the author introduced several land cover flow classification systems that they did not use before describing the system that they did use. I would recommend removing the discussion of the unused systems, and just focus on describing the system that the authors did use.
Page 7, Line 285: It is unclear to me exactly what Figure 2 is showing. Is LCF 2 a sub-flow of LCF 22, or is it a major land use process that LCF 22 has been grouped into? Please add a few sentences in the preceding paragraph explaining the process for classifying flows according to processes.
Page 8, Line 288-328: I think that this section has too much unnecessary information that distracts from the authors' main point. I would recommend removing the discussion of the indices that the authors did not use, and just focus on the ones that they did use.
Results:
Page 9, Line 350: It looks like many of the CLC categories listed in the legend of Figure 3 (discontinuous urban fabric, water bodies, etc) were not present in the study area in either 1990 or 2018. Is this true, or are they simply not visible at the scale of the map?
Page 10, Line 385: I would recommend removing the comma after "conclude".
Author Response
Reviewer 2
There was a lot of grammatical or typographical suggestions – all of them were accepted and are highlighted yellow without explanaiton. All other content suggestions and our revisions are explained below and are highlighted green.
„I think that this paper makes a valuable contribution to environmental monitoring in Slovakia, but substantial grammatical revisions are necessary to prepare it for publication. My specific suggestions are outlined below.“
Abstract:
Page 1, Line 20: I would recommend adding "in the Slovak Republic" after "Low Tatras National Park", as many readers may not know where the Low Tatras National Park is located. → revised
Page 1, Line 20: I would recommend removing the word "the" before "Corin Land Cover". → revised
Page 1, Line 20: I think that "Corin Land Cover" should be changed to "Corine Land Cover". → revised
Introduction:
Page 1, Line 37: I would recommend removing the comma between "Asia" and "South America" and adding the word "and". → revised
Page 1, Line 38: I would recommend removing the "as well" after "Slovakia". → revised
Page 1, Line 40: I would recommend changing "their leading causes" to "the leading causes of land cover changes". → revised
Page 1, Line 42: I would recommend changing "had been" to "was". → revised
Page 2, Line 53-56: This is more a matter of personal taste, so please feel free to disregard: I found the "On the one hand ... on the other hand" structure of this sentence confusing. I would recommend changing the sentence to something more like the following: "One prominent body of research on the mountain environment in Slovakia focuses on geodynamic and geomorphological processes. A second prominent body of research focuses on changes in the landscape structure..." → revised
Page 2, Line 62: Again, this is more a matter of personal taste, but I think the phrase "significantly essential" is redundant. I would recommend removing "significantly". → revised
Page 2, Line 72: The CLC is considered to be the "most complex" what? Please specify. → The sentence was modified → The monitoring of landscape changes in Europe is mostly conducted using the Corine Land Cover (CLC) database [22, 7, 23, 24, 10], which is considered to be the most complex database of spatial-temporal data.
Page 2, Line 76: I think that "valuable" may not be the most appropriate word choice here. Would "accurate" or "complete" or "sophisticated" be more appropriate to the author's meaning? → The sentence was modified → The CLC database of Slovakia is one of the most accurate and complete.
Page 2, Line 83-84: How did the authors decide upon these time periods? Was it based on data availability or historical context? → Explanaiton was added to the line 85. → The main aim of our research is to evaluate land cover changes in the Low Tatras National Park between 1990-2018. Selection of time periods was primarily based on data availability. These data also capture significant historical context since social-political changes in 1989, accession to European Union in 2004 to nowadays.
Materials and Methods:
Page 3, Line 109: I would recommend specifying in the caption of Figure 1 that the highlighted region in the figure is the Slovak Republic. → „in Slovakia“ was added to the caption of the Figure
Page 3, Line 119 - Page 5, Line 177: Since the focus of this paper is on land cover changes, I don't think so much detail about the geology, climate, rainfall, etc of the park is necessary. I would recommending revising this section to make it shorter and more streamlined. → The section about study area was shortened and streamlined.
Page 6, Line 246 - Page 7, Line 282: I found it a little bit confusing that the author introduced several land cover flow classification systems that they did not use before describing the system that they did use. I would recommend removing the discussion of the unused systems, and just focus on describing the system that the authors did use. → Definition of other land cover flows classifications was removed. They are just mentioned in references.
Page 7, Line 285: It is unclear to me exactly what Figure 2 is showing. Is LCF 2 a sub-flow of LCF 22, or is it a major land use process that LCF 22 has been grouped into? Please add a few sentences in the preceding paragraph explaining the process for classifying flows according to processes. → Explanation was added above the picture. We have focused on the main classes of land cover flows (e.g. LCF7, which consists of sub-classes LCF71, LCF72, LCF73, LCF74). Picture with LCF methodology was modified, too.
Page 8, Line 288-328: I think that this section has too much unnecessary information that distracts from the authors' main point. I would recommend removing the discussion of the indices that the authors did not use, and just focus on the ones that they did use. → All unnecessary information was removed.
Results:
Page 9, Line 350: It looks like many of the CLC categories listed in the legend of Figure 3 (discontinuous urban fabric, water bodies, etc) were not present in the study area in either 1990 or 2018. Is this true, or are they simply not visible at the scale of the map?→ There are some small areas near the border of national park and they are not so visible. They are visible on zoom-in picture below.
Page 10, Line 385: I would recommend removing the comma after "conclude". → revised
